# Participant Recruitment Issues in Child and Adolescent Psychiatry Clinical Trials with a Focus on Prevention Programs: A Meta-Analytic Review of the Literature

**DOI:** 10.3390/jcm12062307

**Published:** 2023-03-16

**Authors:** Deniz Kilicel, Franco De Crescenzo, Giuseppe Pontrelli, Marco Armando

**Affiliations:** 1Developmental Imaging and Psychopathology Lab, Department of Psychiatry, School of Medicine, University of Geneva, 1205 Geneva, Switzerland; 2Department of Psychiatry, University of Oxford, Oxford OX1 2JD, UK; 3Pediatric University Hospital-Department (DPUO), Bambino Gesù Children’s Hospital, 00165 Rome, Italy; 4Unité D’Hospitalisation Psychiatrique Pour Adolescents, Department of Psychiatry, Centre Hospitalier Universitaire Vaudois and University of Lausanne, University Service for Child and Adolescent Psychiatry, Avenue Pierre Decker 5, 1011 Lausanne, Switzerland

**Keywords:** recruitment, clinical trial, psychiatry, child and adolescent

## Abstract

**Introduction**: There is a strong need to conduct rigorous and robust trials for children and adolescents in mental health settings. One of the main barriers to meeting this requirement is the poor recruitment rate. Effective recruitment strategies are crucial for the success of a clinical trial, and therefore, we reviewed recruitment strategies in clinical trials on children and adolescents in mental health with a focus on prevention programs. **Methods**: We reviewed the literature by searching *PubMed*/*Medline*, the *Cochrane Library* database, and *Web of Science* through December 2022 as well as the reference lists of relevant articles. We included only studies describing recruitment strategies for pediatric clinical trials in mental health settings and extracted data on recruitment and completion rates. **Results**: The search yielded 13 studies that enrolled a total of 14,452 participants. Overall, studies mainly used social networks or clinical settings to recruit participants. Half of the studies used only one recruitment method. Using multiple recruitment methods (56.6%, 95%CI: 24.5–86.0) resulted in higher recruitment. The use of monetary incentives (47.0%, 95%CI: 24.6–70.0) enhanced the recruitment rate but not significantly (32.6%, 95%CI: 15.7–52.1). All types of recruitment methods showed high completion rates (82.9%, 95%CI: 61.7–97.5) even though prevention programs showed the smallest recruitment rate (76.1%, 95%CI: 50.9–94.4). **Conclusions**: Pediatric mental health clinical trials face many difficulties in recruitment. We found that these trials could benefit from faster and more efficient recruitment of participants when more than one method is implemented. Social networks can be helpful where ethically possible. We hope the description of these strategies will help foster innovation in recruitment for pediatric studies in mental health.

## 1. Introduction

Clinical trials are the main evidence source when assessing therapeutic interventions’ effectiveness, from prevention programs to treatments. To draw reliable conclusions and have significant legitimacy, clinical trials must be conducted on sufficiently large samples. Nevertheless, recruitment difficulties still represent one of the main issues when planning a clinical trial [1,2]. Indeed, 86% of clinical trials display delays due to recruitment difficulties [3,4]. Moreover, 19% of registered trials are either interrupted or suspended because of a flaw in recruitment strategies [5]. As a result, on average clinical trials are costly for funders and discouraging for clinicians, researchers, and participants.

Indeed, different variables can have an impact on the efficacy of recruitment strategies in clinical trials. On the one hand, clinicians often report difficulties in understanding the study’s goal and inclusion criteria, leading them to feel uncomfortable when presenting the study to potential participants. Clinicians also report the tendency not to present the study because they assume patients would not want to participate. Additional workload and lack of interest in the study are other issues that have been reported by clinicians as among the causes of recruitment difficulties [1]. Another barrier for clinicians involved in interventional trials concerns the knowledge that a group of patients will not receive an active treatment. This situation may be experienced as unfair by clinicians, reducing their motivation to engage in the trial [6]. All those reasons can lead clinicians not to promote the study and even to feel uncomfortable regarding their patients’ participation.

On the other hand, psychiatric service users are often seen as more vulnerable than others [7]. For example, their ability to provide informed consent can represent a major issue [8]. Moreover, service users can be undergoing multiple, complex pharmacological treatments with counter-indications to participation in the trial. They may accept to participate in a study for non-optimal reasons or only to avoid being seen as vulnerable by the environment [8,9]. Together, these elements make recruitment for psychiatric clinical trials very difficult. [10]. Aside from those barriers, more obstacles arise when working with children and adolescent clinical populations. Indeed, children and adolescents are considered as a vulnerable population, which generates a number of specific requirements in terms of recruitment strategies [11].

This becomes even more challenging when considering children and adolescents with mental health issues [11,12]. Childhood and adolescence can represent a pivotal moment in the trajectories of developmental psychopathology. The transition from childhood to adolescence is also undoubtedly a period of vulnerability [13,14,15,16], both from a developmental and an ethical point of view. In ethical terms, children and adolescents are vulnerable because their interests are not as fully represented in society as those of adults, their needs tend to be neglected, and their rights are prone to being violated [17,18].

In addition, researchers have to overcome more obstacles such as the under-representation of some ethnic groups and parents’ involvement in the study since in many countries, they are required to provide informed consent on behalf of the minors [19]. Moreover, patients, parents, and caregivers, similar to clinicians, often share concerns regarding the treatment differences between intervention and control groups [6] and possible adverse effects [20,21]. They also recognize their lack of awareness regarding how clinical trials are conducted [20].

This difficulty in developing effective recruitment strategies has two significant clinical implications. Firstly, it may lead to a loss of motivation for conducting clinical trials. Second, the clinical trials conducted may not have sufficient statistical strength to attain reliable conclusions about the effectiveness of a certain therapeutic intervention. This is especially true for trials dedicated to early intervention.

This study aims to review and synthesize the current evidence on recruitment strategies in child and adolescent psychiatry prevention/intervention clinical trials in order to be able to understand which techniques are most effective in dealing with the difficulties outlined above.

## 2. Methods

### 2.1. Search Strategy

This search strategy was performed on *PubMed/Medline*, the *Cochrane Library* database, and *Web of Science* in December 2022, using different terms in relation to clinical trials on children and adolescents with mental health issues focusing on patient recruitment. For more details, search details and a flow-chart can be found in Table 1 and Figure 1.

### 2.2. Study Selection Criteria

Two researchers (D.K. and F.D.C.) independently screened titles and abstracts from all databases for relevance. Two researchers (D.K. and F.D.C.) retrieved and assessed the full texts to determine eligibility. Disagreements were resolved by consensus with another researcher (M.A.). We selected only studies focused on clinical trials in mental health for patients aged 0–18 years that provided information about study recruitment strategies and efficacy.

### 2.3. Data Extraction and Analyses

Multiple data were extracted from the final list of studies by one researcher (D.K.) and verified by another researcher (F.D.C.). These included target population, recruitment methods and incentives adopted, recruitment period, study length and patient involvement, and number of participants.

Our outcomes included the following:(1)Reached participants, defined as all subjects with whom a first contact was made;(2)Recruitment rate, defined as the number of people who accepted to participate out of those to whom researchers reached out;(3)Completion rate, defined as the number of participants who finished the study with all timepoints available over the number of recruited.

After data extraction, we performed meta-analyses of proportion, adopting an approximate likelihood approach by using the Freeman–Tukey double arcsine transformation for computation of the pooled estimates and performing back transformation for stabilizing variances [22]. All the studies were retained independently of extreme proportions (0% and 100%). Descriptive subgroups analysis was performed. Analyses were conducted using the multiple grouping variables listed below:
Recruitment and completion rate in terms of:
○Participant characteristics: (1) common disorders, (2) prevention programs, and (3) rare disease;○Presence or absence of monetary incentive;○Recruitment method used: (1) clinical settings, (2) community setting, (3) social network, and comparison between the use of only one or multiple methods.



All analyses were conducted on Stata, version 16.1. Two authors (D.K., F.D.C.) independently assessed the quality of each study and potential sources of bias based on the Newcastle–Ottawa scale, scored as the percentage of items considered to be of high quality among the scale’s characteristics (quality assessment available in Appendix A). Disagreements were resolved by consensus or with the help of a third reviewer (M.A.).

## 3. Results

Our search yielded a total of 16,328 articles before removal of duplicates. From those, 107 were considered eligible and subjected to full-text review (see PRISMA flow chart, available in Figure 1). In total, 13 studies from 21 publications met our inclusion criteria and were included (the complete list of studies can be found in Appendix A).

In total, 14,452 participants were enrolled in the 13 studies, ranging from 1 to 11,312, with a median of 103 participants. The proportion of males in samples varied from 0 to 58%, as some studies focused on gender-specific topics (median = 41.9%). Detailed characteristics of included studies are described in Table 2.

### 3.1. Recruitment Strategies and Promotion

Three clusters of recruitment methods emerged: (1) the community setting, including word of mouth, schools, and community organizations or structures; (2) clinical setting, which involves recruitment on a medical site where researchers are present at an outpatient clinic, where clinicians themselves act as gatekeepers and introduce the study, or the recruitment is integrated into the usual health check; (3) recruitment through social network, ranging from remote methods (phone, email, or postal mail), to newspapers, online (webpage, Google, TV, and radio,) and social media (Facebook, Twitter, and YouTube). Details on the recruitment strategies used are summarized in Appendix A. Seven studies used clinical methods and/or social network, whereas community setting was engaged in six studies.

Most studies used only one recruitment method (7/13), four studies used two methods, and two used all three identified types of methods (see Appendix A). Most of the studies using only one recruitment method (5/8) used solely clinical settings. When combining two methods, authors mostly used community and social networks together, leading to a larger outreach, especially to a typically developing population (3/4).

Recruitment periods lasted between 12 and 182 weeks (median = 54 weeks). Four studies did not report this information [22,25,27,32,34]. For more details on recruitment promotion, please see Appendix A.

Recruitment efficiency and participant retention.

#### 3.1.1. Recruitment Rate

As reflected in Appendix A, between 1.2 and 79.8% of people were recruited after being invited to participate in the study. The information could not be found for two studies. The highest participation rate appeared to be in a study using all three types of recruitment strategies.

The overall recruitment rate of included studies (information available for only 11/13) was 37.7% (95%CI: 24.7–51.6). The subgroup analysis (see Appendix A) did not show group differences between common disorders (40.5%, 95%CI: 12.0–73.0), prevention programs (35.5%, 95%CI: 18.5–54.6), and rare disease (38.2%, 95% CI: 26.7–50.8).

The use of monetary incentive led to a non-significant but higher recruitment rate of 47.0% (95%CI: 24.6–70.0) compared to studies not using monetary incentives (32.6%, 95%CI: 15.7–52.1; see Appendix A).

According to the selected studies, using multiple recruitment methods enhanced the number of people reached. Social and community settings reached more people in a shorter period compared to clinical settings (Figure 2). Indeed, studies using more than one method had a higher but non-significant recruitment rate (56.6%, 95%CI: 24.5–86.0) compared to studies using only one regardless of the methods used (23.8%, 95CI: 10.8–39.8) (see Appendix A). Overall, using only clinical recruitment sites led to having a negative impact on consent signing, with only 13.7% (95%CI: 0.0–49.2) of reached people consenting to take part in the study. One study showed that combining clinical recruitment sites with social networks improved recruitment to 72.2% (95%CI: 66.8–77.2).

Two effective recruitment methods appeared to be the use of social (42.4%, 95%CI: 40.1–44.6) and community + social networks (46.5%, 95%CI: 40.3–52.7). In one study, we found that community setting alone showed a smaller recruitment rate of 25.9% (95%CI: 23.3–28.7). When all three methods were combined, the recruitment rate dropped to 17.7% (95%CI: 16.3–19.1). Of note, the social network used in the last two studies consisted of flyers or newspaper advertisements and not online social networks. For detailed information, see Appendix A.

#### 3.1.2. Completion Rate

The average completion rate was 82.9% (95%CI: 61.7–97.5; please see Appendix A for more details). Nine out of twelve studies showed retention rates above 75%. When looking at the subgroup analyses on the target population, rare disease (96.2%, 95%IC: 80.4–99.9) and common disorders (88.2%, 95%CI: 47.8–100.0) showed higher completion rates on average compared to prevention programs leading to smaller completion rates (76.1%, 95%CI: 50.9–94.4). Of note, some individual studies showed broad confidence intervals due to the very small sample size.

Monetary incentives again lead to a higher completion rate (91%, 95%CI: 72.1–100.0) on average compared to lack of monetary incentive (76.3%, 95%CI: 44.0–98.4; see Appendix A).

Recruitment methods used also led, on average, to different completion rates. The most fruitful recruitment methods were social (99.5%, 95%CI: 98.7–100.0) and community (99.6%, 95%CI: 98.0–100.0) networks, followed by the use of both (84.1%; 95%CI: 76.7–90.4). Using clinical and social recruitment methods also resulted in a high completion rate (81.7%, 95%CI: 75.9–86.6). Recruiting participants only at clinical settings (64.0%, 95%CI: 6.8–100.00) or using all three methods together (47.2%; 95%CI: 43.0–51.5) had the lowest completion rates (see Appendix A).

The number of recruitment methods used did not seem impact the completion rate since it always remained quite high. The mean rate was 88.1% (95%CI: 59.8–100) for any one recruitment method used regardless of the method and 75.6% (95%CI: 52.8–92.8) for multiple methods (see Appendix A). Quality ratings of the studies averaged 68.7% (ranging 54% to 92%) of the maximum attainable score with the Newcastle–Ottawa scale.

## 4. Discussion

In this study, we systematically reviewed the current evidence on recruitment strategies in child and adolescent psychiatry prevention/intervention clinical trials.

### 4.1. Recruitment Methods

Three main different recruitment methods emerged from this review: (1) community settings, including word of mouth, schools, and community-based organizations; (2) clinical settings, either research clinics or clinician referrals; and (3) social networks, encompassing remote methods where the potential participants did not meet anyone from the research team but found the flyer of the study either on paper or through social media, email, or webpages.

Overall, no recruitment method showed a difference in its efficacy, but, at a trend level, the number of methods used improved recruitment, with higher rates when more than one method was used. When using multiple methods, more patients were reached and accepted to participate. The advantages and disadvantages of each method are summarized in Figure 3.

Using only clinical settings for recruitment led to longer recruitment periods with fewer people reached compared to other methods. This low recruitment rate may be explained by the fact that in this case, clinicians are often the promoters of the study. In fact, as we have seen in the studies considered in this review, clinicians often report multiple obstacles such as difficulty in explaining research methods, poor understanding, and priority given to their patients’ well-being [1,36].

In contrast, social network and community settings showed better results in terms of reaching out to potential participants but not necessarily leading to higher recruitment rates. Social media appeared to reach out to far more people than other recruitment methods. With a high growth and expansion rate and a strong prevalence in young people’s daily life [26], it is one of the best ways to attract their attention with a relatively low investment of resources [37]. On the other hand, social media can encounter more difficulties in term of ethical requirements [38] and because of the very fast development of new technologies and platforms. Indeed, by the time researchers receive their ethics agreement, and especially for longitudinal research, the chosen platform is likely to be outdated.

One study using social media was not in line with the others [39] since it displayed a long recruitment period and a low number of people reached, which appeared closer to the results of use of clinical settings. Actually, in this particular study, clinical research centers were acting as gatekeepers. The research team provided supplementary recruitment material described as social media (i.e., leaflets), but the centers were free to use them or not. Thus, the researchers (?) retained their role of study presentation.

### 4.2. Type of Participants

Although all the studies focused on children and adolescents, three clusters of participants were found, and their participation in research was compared: (1) common disorders (i.e., depression, anxiety, and mood disorders), (2) prevention programs targeting typically developing adolescents, and (3) rare disorders, in this case spina bifida. Moreover, almost half of the identified studies (6/13) concerned early intervention or prevention programs. The proportion of people who consented after being contacted did not depend on the clustered target population and was around 40% in all three cases (?). This homogeneity in recruitment rates between groups can be explained by the fact that all three cluster populations were numerous or easily reachable since even rare disorders are usually recorded in a registry. To the best of our knowledge, there is no specific information on expected recruitment rates for this type of trials and population, but the numbers seem in line with the need for extended recruitment periods faced by many researchers [40]. One factor that enhanced recruitment rates was the use of monetary incentives. As previously shown [41], financial incentives can improve recruitment rate by 14%. However, their use is being criticized, as it may lead to participant biases (i.e., people with financial struggles, mostly students, etc.) and can be ethically questionable since people may participate in research for non-ethical reasons or might be prone to being pressured [42].

### 4.3. Discrepancy between Recruited and Completed

In contrast with recruitment rate and in line with existing evidence [43,44], the completion rate was very high for all participant clusters, even though prevention programs showed lower rates on average. This typically developing population, although easy to reach and recruit, might be less motivated compared to young people with clinical disorders [45]. Completion rates were not influenced by the recruitment method used or the presence of incentives. Globally, it appeared that once people were involved in a trial, they most often stayed in it and completed it. The difference between recruitment and completion rates could also be explained by group differences between people who consent, who are already more motivated than those who do not, and people who do not enter the trial at all.

### 4.4. Limitations

This study has several limitations. Firstly, the results shown in this article should be taken cautiously, as they report results that diverge on average but have overlapping confidence intervals.

Secondly, we focused our review on papers that extensively described the recruitment strategy. This approach can lead to a selection bias since it is well known that, globally speaking, researchers are more prone to provide information concerning recruitment strategies when they are successful. Moreover, there is also a tendency not to publish unsuccessful studies, which leads to a lack of information concerning recruitment strategies adopted in those trials [46,47].

Thirdly, we did not find reports on many possible confounding variables such as people’s intrinsic motivation to participate or their parents’ background. Moreover, no quality control was used regarding the selected studies [48].

Another limitation is that while the motivation of patients and relatives was taken into account as a variable, the motivation of researchers to engage in recruitment was not investigated. This clearly represents a limitation that should be resolved in future studies. The motivation of the researcher to invest in the study is in fact a relevant variable with respect to the success of the study itself.

Moreover, the articles included in this review were all either from northern Europe or the USA, where multiple aspects of research such as regulations, ethics, and logistics are already in place, which is often not the case in developing countries [49]. Researchers in low- and middle-income countries (LMIC) might have to face different challenges that we may have not fully discussed.

## 5. Future Perspective

Aside from the traditionally used methods, new recruitment strategies are currently under development. Indeed, sharing clinical data before and during a clinical trial can elicit worries regarding privacy infringement. One emerging and promising method that has not yet been widely implemented is blockchain technology deriving from the financial sector, of which the most popular example is Bitcoin [50]. This information-holding and transaction technology has already proven to be robust, secure, and auditable [51].

Within the perspective of clinical trials, new methods that allow for a trustworthy sharing of information, such as blockchain technology, could be used to identify and recruit registered participants [52].

We also believe that it is very important to track recruitment strategies and rates in a more systematic way. This would allow a higher level of detail and transparency and would facilitate understanding of which strategies work best and which do not. This lack of consistency has already been underlined [53] but, unfortunately, still remains an issue in clinical trials. In this sense, and in line with other authors [2], we strongly recommend using an information-gathering datasheet that could help improve recruitment strategies by comparing them according to specific populations targeted and methods adopted, including within the same study. This information is most often lacking in the reviewed articles (see Appendix A).

## 6. Conclusions

In summary, it seemed adequate to use multiple strategies and sites to recruit more participants. In general, empowering people by giving them access to information allowing them to decide whether or not they will participate worked better than when the information is provided only by clinicians. Monetary incentives also had a positive impact on recruitment, although they should be used with caution. Interestingly, once young people were involved in the studies, they most often completed them.

Since no method appeared to be superior compared to others, researchers must consider alternative ways to improve recruitment. Most studies reach out to a patient population through clinicians acting as gatekeepers. Reducing the workload of presenting a trial by using handout flyers or leaflets can be effective. To help clinicians improve their understanding and presentation of the studies, the research team should be available and remind on-site teams of their presence through recurrent emails, reminders, and ad hoc presentations of the studies’ progress, rational, and preliminary results.

## Figures and Tables

**Figure 1 jcm-12-02307-f001:**
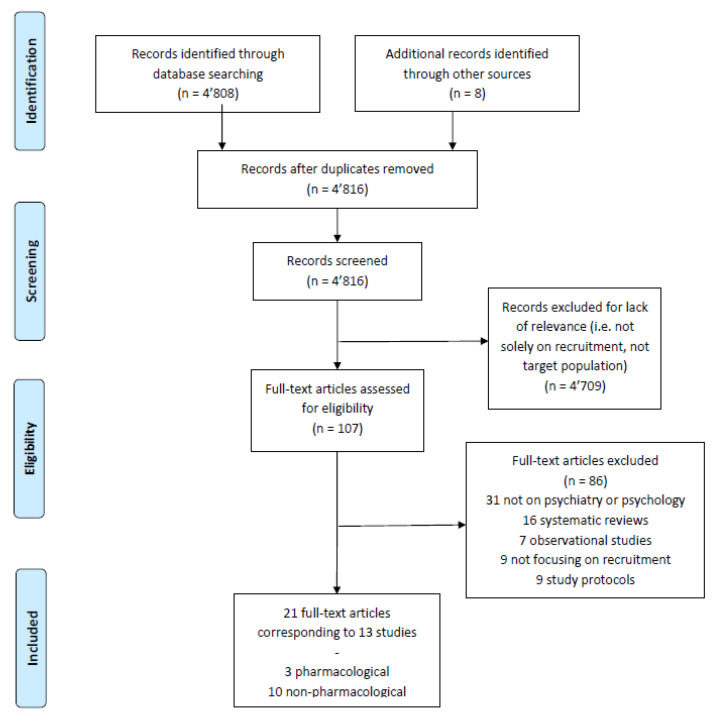
Flowchart of studies included in the review.

**Figure 2 jcm-12-02307-f002:**
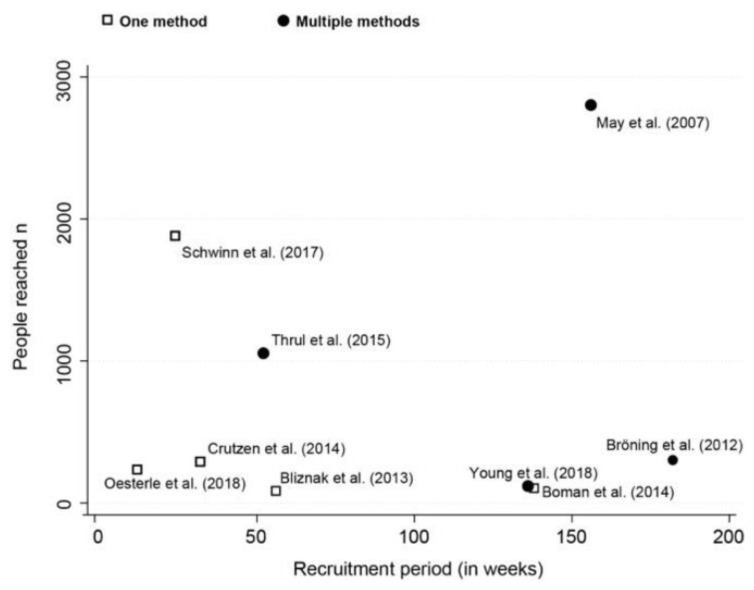
Scatterplot of available studies (n = 9) for recruitment period in weeks and number of people reached (i.e., with whom a first contact was made). Separated according to study recruitment methods [23,24,26,28,29,30,31,33,35].

**Figure 3 jcm-12-02307-f003:**
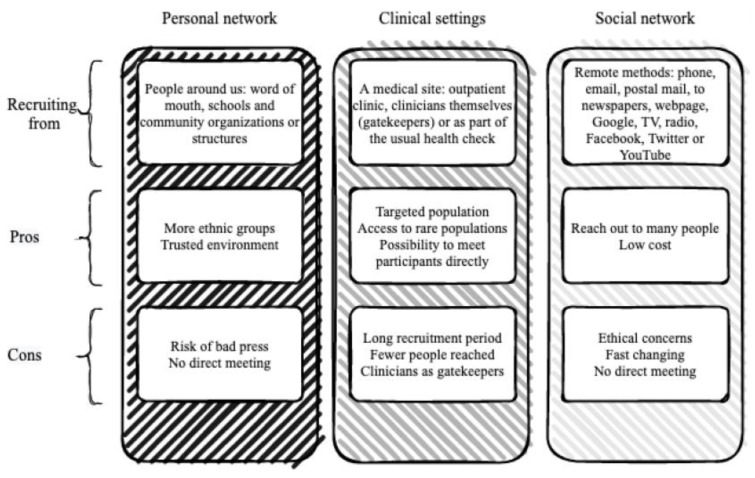
Advantages and disadvantages of using personal network, clinical settings, and social network as recruitment strategies.

**Table 1 jcm-12-02307-t001:** Search Strategy.

	**Search Terms-PubMed**	**N° of Papers**
**#1**	Adolescent[Mesh] OR “Adolescent Medicine”[Mesh] OR Child[Mesh] OR “Minors”[Mesh] OR Pediatrics[Mesh] OR “Young Adult”[Mesh] OR child* OR schoolchild* OR kid OR kids OR toddler* OR adoles* OR teen* OR boy* OR girl* OR minors* OR underag* OR “under age” OR “juvenil*” OR youth* OR kindergar* OR puberty OR pubescen* OR prepubescen* OR prepuberty* OR pediatric* OR paediatric* OR peadiatric* OR preschool* OR schoolage	5,179,448
**#2**	psychiatry OR “mental health” OR psychology OR psychosocial OR social	3,067,902
**#3**	“Patient Selection”[Mesh] OR Enrolment OR “Patient Selection*” OR “Selection, Patient” OR “Selections, Patient” OR “Research Subject Recruitment” OR “Recruitment, Research Subject” OR Recruitments, Research Subject” OR “Research Subject Recruitments” OR “Subject Recruitment, Research” OR “Subject Recruitments, Research” OR “Research Subject Selection” OR “Research Subject Selections” OR “Selection, Research Subject” OR “Selections, Research Subject” OR “Subject Selection, Research” OR “Subject Selections, Research” OR “Selection for Treatment” OR “Selection for Treatments” OR “Treatment, Selection for” OR “Treatments, Selection for” OR “Selection of Subjects” OR “Subjects Selection” OR “Subjects Selections” OR “Patient Recruitment” OR “Patient Recruitments” OR “Recruitment, Patient” OR “Recruitments, Patient “ OR “Selection Criteria”	4724
**#4**	#1 AND #2 AND #3	614
	**Search Terms–Cochrane Library**	**N° of Papers**
**#1**	child* OR schoolchild* OR kid OR kids OR toddler* OR adoles* OR teen* OR boy* OR girl* OR minors* OR underag* OR “under age” OR “juvenil*” OR youth* OR kindergar* OR puberty OR pubescen* OR prepubescen* OR prepuberty* OR pediatric* OR paediatric* OR peadiatric* OR preschool* OR schoolage	327,615
**#2**	psychiatry OR “mental health” OR psychology OR psychosocial OR social	194,660
**#3**	Enrollment OR “Patient Selection*” OR “Research Subject Recruitment” OR “Research Subject Recruitments” OR “Research Subject Selection” OR “Research Subject Selections” OR “Selection for Treatment” OR “Selection for Treatments” OR “Selection of Subjects” OR “Subjects Selection” OR “Subjects Selections” OR “Patient Recruitment” OR “Patient Recruitments” OR “Selection Criteria”	55,601
**#4**	#1 AND #2 AND #3	4194
	**Search Terms–Web of Science**	**N° of Papers**
**#1**	child* OR schoolchild* OR kid OR kids OR toddler* OR adoles* OR teen* OR boy* OR girl* OR minors* OR underag* OR “under age” OR “juvenil*” OR youth* OR kindergar* OR puberty OR pubescen* OR prepubescen* OR prepuberty* OR pediatric* OR paediatric* OR peadiatric* OR preschool* OR schoolage	7,236,284
**#2**	psychiatry OR “mental health” OR psychology OR psychosocial OR social	6,527,962
**#3**	Enrollment OR “Patient Selection*” OR “Research Subject Recruitment” OR “Research Subject Recruitments” OR “Research Subject Selection” OR “Research Subject Selections” OR “Selection for Treatment” OR “Selection for Treatments” OR “Selection of Subjects” OR “Subjects Selection” OR “Subjects Selections” OR “Patient Recruitment” OR “Patient Recruitments” OR “Selection Criteria”	257,535
**#4**	#1 AND #2 AND #3	11,522

* MeSH: Medical Subject Heading.

**Table 2 jcm-12-02307-t002:** Characteristics of included studies to improve recruitment for children and adolescents in mental health.

Author (Year)	Origin	Study Design	Recruited Population	Participants n	Age Mean (SD)	Age Range	Male %	Monetary Incentives
Bliznak et al. (2013) [23]	Germany	RCT	Children and adolescents with depression	1	14.0 (0)	7 to 17	0.0	No
Boman et al. (2014) [24]	Sweden	RCT	Adolescents with anxiety	55	15.2 (1.80)	12 to 19	58.0	No
Breland-Noble et al. (2012) [25]	United-States	RCT	Adolescents with depression	16	15.1 (1.88)	11 to 17	31.2	No
Bröning et al. (2012) [26]	Germany	RCT	Children of substance abusing parent(s)	218	9.79 (1.87)	8 to 12	52.3	No
Cheung et al. (2017) [27]	Netherlands	Open Label trial	Typically developing adolescents	11 312 ^a^	14.24 (1.13) ^a^	13 to 16	47.8	No
Crutzen et al. (2014) [28]	Netherlands	RCT	Typically developing adolescents	12	NA	13 to 14	NA	No
May et al. (2007) [29]	United-States	RCT	Adolescents with depression	439	14.6 (1.50)	12 to 17	45.6	No
Oesterle et al. (2018) [30]	United-States	Cohort	Parents of children and adolescents interested in drug use prevention	103	NA	11 to 13 ^c^	1.9	Yes
Schwinn et al. (2017) [31]	United-States	Pre- post-intervention	Typically developing adolescents, drug use prevention	788	13.7 (0.67)	13 to14	0.0	No
Smith et al. (2015) [32]	United-States	RCT	Adolescents with Spina bifida	25	NA	14 to 20	36.0	Yes
Thrul et al. (2015) [33]	Germany	n-RCT	Smoking adolescents	1054	14.8 (1.23)	11 to 19	52.5	Yes
Wagner et al. (2012) [34]	United-States	RCT	Adolescents with depression	334	15.9 (1.60)	12 to 18	30.2	No
Young et al. (2018) [35]	United-States	RCT	Children and adolescents with primary mood disorder	95	11.3 (2.20)	7 to 14	56.8	Yes

NA, not available; n-RCT, non-randomized controlled trial; RCT, randomized controlled trial; ^a^ merged populations; ^c^ age of the children.

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
