# Peer review of "Participant Recruitment Issues in Child and Adolescent Psychiatry Clinical Trials with a Focus on Prevention Programs: A Meta-Analytic Review of the Literature"

_jcm, 2023, doi:10.3390/jcm12062307_

Round 1
Reviewer 1 Report (Previous Reviewer 2)
I would suggest describing the search strategy and flowchart in the main body of the manuscript, not in the appendix.
The Author do not differentiate between studies sponsored by industry companies and studies being just a scientific research.
The authors mention that parents' and patients' motivation is an important issue. What about researchers' motivation (e.g financial, scientific, both)? I believe that that kind of information may be difficult to extract. If so, it should be mentioned in the limitation section.
I would sugegst underlining in the limitation section the scarcity of studies discussing the recruitment strategies and thus a numerous not enough explored issues.
Author Response
We would like to thank the reviewers for their feedback that enabled us to improve the manuscript.
Reviewer 1
Reviewer’s comment 1. I would suggest describing the search strategy and flowchart in the main body of the manuscript, not in the appendix.
Author’s response 1. We thank the reviewer for this suggestion. The table on search strategy and the flow chart have now been added to the main text.
Reviewer’s comment 2. The Author do not differentiate between studies sponsored by industry companies and studies being just a scientific research.
Author’s response 2. To the best of our knowledge, none of the included studies were sponsored by industry companies.
Reviewer’s comment 3. The authors mention that parents' and patients' motivation is an important issue. What about researchers' motivation (e.g financial, scientific, both)? I believe that that kind of information may be difficult to extract. If so, it should be mentioned in the limitation section. I would suggest underlining in the limitation section the scarcity of studies discussing the recruitment strategies and thus a numerous not enough explored issues.
Author’s response 3. We thank the reviewer for this relevant comment. We have now added the following text in the limitations section “Another limitation is that while the motivation of patients and relatives was taken into account as a variable, the motivation of researchers to engage in recruitment was not investigated. This clearly represents a limitation that should be resolved in future studies. The motivation of the researcher to invest in the study is in fact a relevant variable with respect to the success of the study itself.”
Reviewer 2 Report (Previous Reviewer 1)
The method section must describe the METHOD and instrument used for quality assessment. How have authors conducted the quality assessment? How many authors? How much disagreement was there and how did authors deal with disagreements? The results section must present the findings of the quality assessment. Authors are advised to check PRISMA=based systematic reviews regarding the reporting of the quality assessment. .
Author Response
We would like to thank the reviewers for their feedback that enabled us to improve the manuscript.
Reviewer 2
Reviewer’s comment 1. The method section must describe the METHOD and instrument used for quality assessment. How have authors conducted the quality assessment? How many authors? How much disagreement was there and how did authors deal with disagreements?
Author’s response 1. We thank the reviewer for the comments which allows us to clarify this issue. We modified the text as following: “Two authors (DK, FDC) independently assessed the quality of each study and potential sources of bias, based on the Newcastle-Ottawa Scale scored as the percentage of items considered to be of high quality among the scale’s characteristics (quality assessment available in appendix 15). Disagreements were resolved by consensus or with the help of a third reviewer (MA).”
Reviewer’s comment 2. The results section must present the findings of the quality assessment. Authors are advised to check PRISMA=based systematic reviews regarding the reporting of the quality assessment.
Author’s response 1. We added the following to the text: “Quality ratings of the studies averaged 68.7% (ranging 54% to 92%) of the maximum attainable score with the Newcastle-Ottawa Scale.”
This manuscript is a resubmission of an earlier submission. The following is a list of the peer review reports and author responses from that submission.
Round 1
Reviewer 1 Report
The review aimed to review the literature on recruitment strategies and recruitment-related issues in child and adolescent clinical trials. The review included 13 studies. Findings indicated the benefits of using multiple recruitment methods, including social networks. Findings of the review may inform the design of future trials, specifically with regards to recruitment and sampling.
Overall, I found it an important topic, and the findings certainly are potentially interesting for future trials. Nonetheless, I have a few comments, including crucial concerns regarding the methodology. I hope that these comments, detailed below, may contribute to improving the manuscript.
Title
Maybe authors can consider naming it ‘a meta-analytic review of the literature’, as that would reflect the methodology of the review.
Introduction
The review is focused on clinical trials with children and adolescents. However, only six lines on page 2 are addressing this population (“aside from those barriers … “), and it is not clear how the rest of the introduction applies to this population. As such, the introduction does not provide a strong rationale for this review.
Several sentences in the introduction are not clear. For example:
Page 1: “ … meet the needs of participants”. … to recruit the required number of participants?
Page 2: “ … clinical trials on the way those are conducted.” … regarding how clinical trials are conducted.
There are other examples throughout the manuscript. Thorough editing, including for writing in English is recommended.
Methods
Overall, I recommend conducting the review according to PRISMA guidelines. In fact, it is not clear why authors have not adhered to these guidelines, as it would reassure the readers of the quality of the review.
The search was conducted in one database. This is very unusual for a systematic review, as it introduces a bias inherent to the selected database. It is recommended to use additional databases, for example, Embase, Emcare, Scopus, Web of Science …
The search was conducted in July 2021, i.e., one year and four months ago. It is a good practice to submit a systematic review within six months of the searches. As such, the review would include recent studies and the manuscript will not be outdated by the time it gets published. It is not clear why the search was not updated prior to submitting the manuscript.
Study selection criteria
What was the definition of children and adolescents? What age ranges were included in the review?
Study selection
Usually, one researcher will conduct the searches, delete the doubles, and make a first section based on titles and abstract.
Next, two researchers independently make a further selection based on the full texts. Disagreements can be solved by discussion or by involvement of a third researcher. Interrater reliability should be reported.
Such a systematic approach (also regarding data extraction, below) is required to cater for, amongst other things, researcher bias.
Quality assessment of the included studies would provide important information for the readers as it indicates how reliable methods and findings of the included studies are. It is not clear why quality assessment was not conducted.
Data extraction
Usually, two researchers independently extract the data. Again, disagreements can be solved by discussion or by involvement of a third researcher. Interrater reliability should be reported.
The many methodological flaws may have had an impact on the findings of the review. Thus, I will refrain from providing comments on Results and Discussion. Nonetheless, the discussion should discuss the study findings in the context of the child and adolescent literature, and I cannot locate many references regarding this population in the discussion section. Thus, I would encourage the authors to check the literature.
Upgrading the methodology of the review will constitute a substantial and essential improvement of the manuscript, and I wish the authors good luck with the revision.
Reviewer 2 Report
Very interesting topic and well-conducted study.
I would suggest adding the more detailed description as well as the flow chart to the body of the manuscript.
Round 2
Reviewer 1 Report
Thank you for the revised version of the manuscript. Though the manuscript has improved, there remain important concerns.
Replying to a previous comment, authors stated that they did not want to conduct a quality assessment of the selected studies, as this would be 'misleading'. I do nut understand this reply. A quality assessment would provide additional information so the reader understands the quality of the included studies. In line with PRISMA guidelines and good practice in systematic reviews, I recommend that authors conduct a quality assessment.
Replying to another comment, authors have now conducted searches in two databases instead of one database. However, this is still very minimal and rather poor for a systematic review. I see no reason why authors would not conduct their searches in multiple health-related databases. The quality of the systematic review will improve, and it may yield additional eligible studies.
The abstract stated that 13 studies were included. However, the various appendices report on 11 or 12 studies. The sentence: "Most studies used only one recruitment method (8/14)" suggests that 14 studies were included. Please correct or clarify to avoid confusion in the readers.
The first sentence in section 2.2 is not English. Also, the Discussion and Conclusion sections (both 'new and 'old' sentences') are written in poor English. As mentioned before, thorough editing by a native English speaker/researcher is recommended.
Good luck with the revision of the manuscript.